# Prediction of Metabolic Profiles from Transcriptomics Data in Human Cancer Cell Lines

**DOI:** 10.3390/ijms23073867

**Published:** 2022-03-31

**Authors:** Maria Vittoria Cavicchioli, Mariangela Santorsola, Nicola Balboni, Daniele Mercatelli, Federico Manuel Giorgi

**Affiliations:** 1Department of Pharmacy and Biotechnology, University of Bologna, 40126 Bologna, Italy; maria.cavicchioli@studenti.unitn.it (M.V.C.); mariangel.santorsola@unibo.it (M.S.); nicola.balboni@unibo.it (N.B.); daniele.mercatelli2@unibo.it (D.M.); 2Department of Information Engineering and Computer Science, University of Trento, 38123 Trento, Italy; 3Department of Biology, University of Pisa, 56126 Pisa, Italy

**Keywords:** metabolomics, transcriptomics, machine learning, correlation networks, cancer

## Abstract

The Metabolome and Transcriptome are mutually communicating within cancer cells, and this interplay is translated into the existence of quantifiable correlation structures between gene expression and metabolite abundance levels. Studying these correlations could provide a novel venue of understanding cancer and the discovery of novel biomarkers and pharmacological strategies, as well as laying the foundation for the prediction of metabolite quantities by leveraging information from the more widespread transcriptomics data. In the current paper, we investigate the correlation between gene expression and metabolite levels in the Cancer Cell Line Encyclopedia dataset, building a direct correlation network between the two molecular ensembles. We show that a metabolite/transcript correlation network can be used to predict metabolite levels in different samples and datasets, such as the NCI-60 cancer cell line dataset, both on a sample-by-sample basis and in differential contrasts. We also show that metabolite levels can be predicted in principle on any sample and dataset for which transcriptomics data are available, such as the Cancer Genome Atlas (TCGA).

## 1. Introduction

Cancer is a group of histologically heterogeneous diseases, both across patients and within the tumor itself [1]. The root of this diversity has a molecular basis, observable at the genomic level in the form of different genomic alterations (somatic mutations and copy number alterations) and various epigenetic and transcriptional states [2]. While many successful efforts have brought fruit in identifying clinically relevant cancer subtypes [3], for most tumor types, this heterogeneity is poorly understood, translating into not fully predictable disease progressions and varying responses to therapeutic strategies [4]. Several efforts at classifying cancers on the basis of genomic statuses, such as the Cancer Genome Atlas (TCGA) study of 2014 [5], were based on the presence/absence of Epstein–Barr viral infection, and secondarily on the genomic status of tumors, but ultimately failed at inferring clear predictors for disease progression and treatment [6]. Moreover, cancer heterogeneity is also evident within the tumor itself, which appears as a complex microenvironment comprising several cytological and histological types, such as fibroblasts, immune infiltrates, bone marrow-derived cells, and stem cells [7,8].

One source of molecular variability still largely under-investigated in human cancer is the metabolome, i.e., the collection of small molecules, or metabolites, that constitute both the building blocks and energy currency of all living cells, as well as the means to increase biomass [9]. Specific metabolites are highly needed by tumor demands for growth, and the deregulation of metabolic pathways has been defined as one of the key components, or hallmarks, of cancer [10]. Metabolites have been linked to tumor progression: for example, an increase in proline abundance has been directly linked to increased cell plasticity, epithelial-to-mesenchymal transition, uncontrolled proliferation and metastatic progression of cancer cells [11]. Methionine deprivation, on the other hand, has been linked to a decrease in tumor growth [12], possibly via lowering polyamine synthesis, a key pathway for cancer cell proliferation [13].

Beyond the classification of cancer and the understanding of how metabolites act in tumor initiation and progression [14], focusing on the metabolome has also potential benefits for discovering new treatments, as it is intrinsically more accessible than the other -omes (genome/transcriptome/proteome/epigenome) to dietary and pharmacological interventions [15]. Novel anticancer strategies based on metabolite supplements or deprivations may represent the next translational strategy for therapeutic applications on anticancer therapy [16].

Despite its importance, the metabolome has been considerably less studied than the genome and the transcriptome, due to the absence of high-throughput methods as wide, accurate, and affordable as quantitative techniques for nucleic acids, currently mostly represented by next generation sequencing (NGS). There are, however, direct quantifiable interactions existing between metabolites and other cellular components. For example, the metabolome, by supplying acetyl and methyl groups, directly affects the epigenetic modifications of histones and DNA/RNA, and therefore gene expression [17,18]. In addition, the production of metabolites depends on the abundance of enzymes synthesizing them, and therefore of the messenger RNAs (mRNAs) from which such enzymes are translated, one example being the levels of kynurenine significantly correlated with the levels of the enzymes IDO1, IDO2, and RDO, which catalyze the reaction of its precursor N-formylkynurenine [19]. In addition, metabolites can affect protein activity through allosteric interactions, one example being cyclic AMP (cAMP) inducing the activity of CRP transcription factor in *Escherichia coli*, which translates into a strong correlation between cAMP and CRP target genes [20]. Correlations between metabolites and transcripts are also, in some cases, highly conserved across species [21].

This mutual, conserved intertwining between metabolome and transcriptome, resulting in the existence of quantifiable correlations between them, has spurred the question of whether metabolite levels could be predicted from transcript abundance, and vice-versa. Given the low cost of RNA-Seq for transcriptome measurement, the possibility to use it to also predict metabolite quantities in cancer samples carries both economic and scientific value. Early studies in the 2010s have been hindered by the small scale and high noise of early datasets, yielding largely inconclusive results [22,23]. More recently, however, two large scale metabolite/transcript datasets have been generated in human cancer cell lines, by the Cancer Cell Line Encyclopedia (CCLE, 890 samples) [19] and National Cancer Institute (NCI-60, 60 samples) [24] consortia, providing new data on which to derive correlations between metabolites and transcripts. This has been accompanied by recent advances in machine learning techniques [25], whose early applications have allowed for predicting somatic mutations from gene expression profiles in human cancer [26], and also metabolite abundances from transcript quantities in yeast [27].

In this manuscript, we will test if the prediction of metabolite levels can be performed in cancer cell lines based on information contained in gene expression profiles (Figure 1). We will first analyze the CCLE cancer cell line dataset and then test if the metabolite prediction can work on different scenarios, specifically the NCI-60 cancer cell line dataset, and patient-derived samples. In order to do so, we will rely on the construction of network-based structures captivating all direct correlation information between metabolites and transcripts, using the *corto* algorithm [28]. The *corto* algorithm is the latest installment of a long succession of co-occurrence methods that derive direct associations between abundances of molecular species [29], using a combination of correlation coefficients and Data Processing Inequality (DPI) to remove indirect connections [30]. These tools have been successfully applied to reverse-engineer biological networks such as protein–protein interaction networks or regulatory networks [31]. More recently, co-occurrence networks have been applied in inferring the activity of transcription factors both on differential signatures (e.g., mutated vs. wild-type) and on a sample-by-sample basis [32], by virtue of a weighted aggregation of all target genes associated with specific transcription factors. Such approaches have also been successfully applied to infer the presence of specific somatic mutations in cancer, proving the existence of a quantifiable association between transcriptional profiles and other cell properties [26], but they remain untested in the task of metabolite abundance prediction.

## 2. Results

The CCLE cancer cell line dataset used in this study was composed of 1019 samples measuring the expression of 24,274 distinct genes, plus 928 samples measuring the abundance of 225 metabolites, of which 136 polar and 89 non-polar (fatty acids) [19]. The intersection of the two portions of the CCLE dataset provided 898 samples; at the moment of writing, the largest existing dataset of co-measured transcripts and metabolites in human cells.

Co-occurrence analysis of this dataset shows clear correlation patterns between metabolites and transcripts, many of which are based on a direct biochemical basis. First, in order to check the consistency of our study with what was previously reported [19], we showed that the *IDO1*/kynurenine correlation, used as an example in the original CCLE publication [19] is positive (correlation coefficient CC = 0.376) and significant (*p* = 1.43 × 10^−31^) (Figure 2A). *IDO1* in fact codes for indoleamine 2,3-dioxygenase, an enzyme converting tryptophan to N-formyl-kynurenine, which then spontaneously transforms into kynurenine [33]. Our analysis shows more significant relationships: the most significant in the dataset is the correlation between *NNMT* gene expression and 1-methylnicotinamide (Figure 2B) with CC = 0.788 (*p* = 6.29 × 10^−191^). *NNMT* encodes for Nicotinamide N-methyltransferase, the enzyme producing 1-methylnicotinamide by adding a methyl group directly from nicotinamide [34].

We investigated more transcript/metabolite pairs (Appendix A shows the pairs most correlated in the CCLE dataset). The second highest correlated metabolite/transcript pair in the CCLE dataset is N-carbamoyl-beta-alanine/*DPYD*, which possesses a correlation coefficient of 0.692 (*p* = 3.55 × 10^−129^). This correlation relationship can be justified by their co-presence in the pyrimidine catabolism pathway, more specifically in the steps converting uracil into beta-alanine. The *DPYD* gene encodes for Dihydropyrimidine Dehydrogenase, an enzyme catalyzing the reaction of uracil into 5,6-dihydrouracil, which then gets converted directly into N-carbamoyl-beta-alanine [35]. An example not involving enzymes is constituted by the *SLC6A8* gene, encoding for a carrier of creatine: in fact, the correlation between *SLC6A8* and phosphocreatine is the 4th highest in the CCLE dataset (CC = 0.655, *p* = 2.42 × 10^−111^), and the correlation between *SLC6A8* and creatine is the 11th highest (CC = 0.627, *p* = 3.61 × 10^−99^).

After having screened the most significant metabolite/transcript relationships, we decided to construct a full correlation map between metabolites and transcripts in the CCLE dataset, using the *corto* algorithm. This algorithm calculates all metabolite/transcript Pearson correlation coefficients (Figure 1B) and then removes indirect ones through other metabolites by applying DPI to all metabolite–metabolite–transcript triplets, removing the edge with the lowest correlation [28] (Figure 1C). Each edge is then tested in multiple bootstraps of the original dataset to assess its likelihood, which can result in the recovery of edges removed by DPI in the full dataset but supported by at least one bootstrap (Figure 1D). Both correlation coefficient and likelihood scores are stored in order to weight the contribution of each transcript in their metabolite network.

The network derived from the CCLE dataset contains 60,162 edges (i.e., significant connections), 198 metabolites, and 10,385 transcripts, and is available as Appendix A.

### 2.1. CCLE-Based Analyses

Initially, we tested the prediction of individual metabolites using one half of CCLE to train a metabolite/transcript network, and the other half to predict metabolite abundance. During the prediction step, only transcript information was included in the information given to the algorithm. Our results show that, for all 157 metabolites for which at least significantly directly correlating transcripts could be found, all predictions achieved a positive correlation coefficient, expressed as the average of 2000 sampling steps (Figure 2C), choosing every time a different subsetting for the training and prediction halves. This was particularly evident for transcripts such as 1-methylnicotiname (CC = 0.73) and butyrobetaine (CC = 0.506), shown in a single prediction run in Figure 2D,E, respectively. The most significant polar metabolites predicted using the CCLE dataset are shown in Figure 3A, including 1-methlnicotinamide, arginine, fructose-1-phosphate, valine, and pyroglutamic acid. The correlation coefficients are above 0.5 (*p* < 1 × 10^−302^) for the top 10 metabolites, indicating a strong conservation in metabolite/transcript correlation structures for several molecules, which allow for consistent prediction of metabolic abundance.

### 2.2. Cross-Dataset Analyses

In order to test whether the capability to predict metabolites would cross the dataset boundary, we obtained a second cancer cell line dataset from the NCI-60 project [36]. This dataset provides 57 samples from different cell lines, where a total of 280 metabolites and 17,987 genes are measured. The resulting *corto* metabolite/gene network obtained from the NCI-60 dataset contains 6613 genes, 280 metabolites, and 24,383 significant edges, available as Appendix A. We then tested whether the NCI-60 could predict metabolite levels on the CCLE dataset, using *corto* with default parameters. In addition, 34 metabolites from the NCI-60 network possessed at least 10 gene interactors contained in both dataset and could be predicted on the CCLE dataset. In Figure 3B, we show the correlation values between the predicted values (inferred via the *corto* algorithm and the NCI-60 network) and the measured values of these 34 metabolites in the CCLE network. The majority (22 out of 34) of the metabolites showed a significant and positive (*p* < 0.05) Pearson’s correlation coefficient between predicted and measured values, with phenylalanine, methionine, uracil, histidine, leucine, tyrosine, and valine having the highest correlation coefficient, above 0.4. Overall, the predictions are significantly better than expected by chance (one-tail Wilcoxon test *p*-value = 1.92 × 10^−5^).

We continued the analysis by applying the CCLE-based metabolite/gene network (described before and available as Appendix A) on the NCI-60 dataset. We used once again default *corto* parameters, calculating metabolite abundance predictions only for those metabolites who had at least 10 genes measured in both datasets in their network neighborhood (default parameter *minsize = 10*). This provided a prediction for 21 metabolites, 11 of which had a positive and significant correlation between predicted and measured levels (Figure 3C), more than expected by chance (one-tail Wilcoxon test *p*-value *p* = 0.000304). Metabolites well predicted in both tests, with PCCs above 0.4, were methionine, tyrosine, leucine, phenylalanine, and uracil. Examples of sample-by-sample prediction scatterplots from which the overall analysis is based are shown in Figure 3D, where predicted levels of phenylalanine and methionine are compared with experimentally measured levels of the same metabolites in both datasets.

It is worth noting that one metabolite, allantoin, in both tests (NCI-60-based prediction on CCLE data, and CCLE-based prediction on NCI-60 data) showed a significant and *negative* correlation between predicted and measured data, indicating an opposite nature in the network structures and edge correlation signs of the two datasets.

### 2.3. Applications on Patient-Derived Data

Theoretically, all transcriptome-wide data could provide information for network-based metabolite prediction, whether on a sample-by-sample basis, or in differential analysis, even patient-derived data. In order to test this, we set out to test whether cancer cell line-derived networks could be applied in predicting metabolite abundances in different, patient-derived transcriptomics contexts. The first tested scenario we selected is the cancer genome atlas (TCGA), the largest patient cancer dataset, which encompasses more than 30 tumor types and more than 20,000 samples. The TCGA dataset provides, for each sample, several data, that may include gene expression, somatic mutations, DNA methylation, copy number alteration, clinical parameters (including overall survival), and protein abundance [5]; despite this plethora of information, no dataset-wide metabolite measurement has been performed so far, leaving the metabolome outside the range of systems biology analyses.

Using our metabolite/gene network from the CCLE dataset (the largest in terms of samples), we applied via the *corto* algorithm a sample-by-sample inference of metabolite abundance across 14 TCGA datasets (selected as the largest ones providing also patient survival information): BLCA (Bladder Urothelial carcinoma), BRCA (Breast Invasive carcinoma), ESCA (Esophageal carcinoma), HNSC (Head and Neck squamous cell carcinoma), KIRC (Kidney renal clear cell carcinoma), KIRP (Kidney renal papillary cell carcinoma), LUAD (Lung adenocarcinoma), LUSC (Lung squamous cell carcinoma), PRAD (Prostate adenocarcinoma), SARC (Sarcoma), SKCM (Skin Cutaneous melanoma), STAD (Stomach adenocarcinoma), THCA (Thyroid carcinoma), and UCEC (Uterine Corpus Endometrial carcinoma). The inferred pan-cancer metabolite prediction was used to test whether each inferred metabolite could be used as a survival predictor in each dataset, and in fact several metabolites were significantly (*p* < 0.01) associated with survival (Figure 4A), both as “positive marker” (i.e., their higher level associated to worse prognosis, in red), or “negative marker” (i.e., their lower level associated to worse prognosis, in blue). For example, a higher (inferred) abundance of kynurenic acid is associated with poorer prognosis in LUAD (Figure 4B, *p* = 6.9 × 10^−8^), confirming experimental results associating kynurenic acid and tumor invasiveness in non-small cell Lung cancer (which includes Lung adenocarcinoma) [37]. Inferred Kynurenic acid levels are a poor prognostic marker for patient survival in several other tumor types, such as HNSC (Figure 4C, *p* = 2.2 × 10^−5^), confirming a general and widely known role of this metabolite in tumorigenesis and tumor progression [38]. Higher predicted levels of specific metabolites could also be associated with a better prognosis; one such example is acetylcarnitine, whose higher levels are associated with a significantly better outcome in KIRC (Figure 4D, *p* = 2.0 × 10^−7^), but also in KIRP, LUAD, STAD, and TCHA (Figure 4A). Our predicted findings on acetylcarnitine reflect experimental evidence associating decreased urinary levels of this metabolite to an altered metabolic state associated with kidney cancer [39], and more general anticarcinogenic roles of acetylcarnitine in cancer [40,41]. Another example is constituted by phosphocreatine, whose higher predicted levels are correlated with poorer prognosis in melanoma (Figure 4E, *p* = 4.4 × 10^−8^); we could report a very strong correlation between phosphocreatine metabolic levels and gene expression of *SLC6A8* (Appendix A), and both are known to be able to promote cancer progression [42] by promoting the cancer hallmark of energetic dysregulation [10].

Many metabolites are significantly correlated with cancer genes, and this mutual relationship is reflected in the CCLE network as well: Figure 4F shows a subnetwork highlighting the seven metabolites most significantly associated with cancer survival (positively or negatively, by the sum of −log10p_survival_ based on data from Figure 4A) and their significant connections with known cancer genes (from the COSMIC Cancer Gene Census [43]). Kynurenic acid seems to be the hub in the metabolite/gene cancer correlation network inferred from CCLE (Figure 4F), and several of the cancer genes significantly correlated with metabolites include pan-cancer actors such as *MAX*, *ATM*, *PTEN*, and *BRAF*. This strengthens the already widely accepted role of metabolites as molecular co-effectors and potential novel biomarkers of cancer [9].

We then proceeded to test the efficacy of the gene-derived metabolite level predictions in a different patient-derived dataset for which we could obtain metabolome and transcriptome paired data in the same samples. Such a dataset was provided by the work of Shchukina and colleagues [44], who measured human monocyte samples from young and old individuals (referred here as the “aging” dataset). One obstacle to metabolite prediction and prediction testing across datasets is by no doubt constituted by the partially non-overlapping nature of metabolomics datasets: while there is high comparability between transcriptomics datasets, due to the existence of few reference versions both for genome sequences and genome annotation, metabolomics is still lacking full standardization, and in fact only 30 metabolites could be traced in common between the three metabolomics datasets used in this study: CCLE, NCI-60, and Aging (Figure 5A). Despite this, the CCLE-based network could predict the differential old vs. young abundance of the Aging dataset with a significant correlation (PCC = 0.4, *p* = 0.017, Figure 5B). Among others, 1-methylnicotinamide was predicted as increased in monocytes from old individuals when compared to young individuals, a prediction confirmed by experimental data, as well as creatine uracil and myristoilcarnitine.

### 2.4. Further Validation

One potential criticism to the use of transcriptomics data is that transcript abundance does not reflect the actual protein levels, and even less so the levels of protein activity. Proteomics data are however more expensive, less standardized, and not as widespread as Transcriptomics data, thanks to the two-decade effort provided by microarrays first, and then RNA-Seq [45]. Several studies have shown how transcript data can be effectively used to predict protein activity [32]; however, in many cases, it would be more logical to build a metabolite predictor based on protein levels, such as the case of the enzyme NNMT correlating with 1-methylnicotinamide levels (Figure 2B). We could, however, show that, in TCGA samples with both transcript and protein levels measured, the correlation between the two molecules for 187 measured genes was significantly higher than expected by chance (*p* = 4.3 × 10^−137^), confirming that transcripts and proteins levels are correlated, and that transcripts are therefore *bona fide* proxies (in metabolite prediction or otherwise) for protein abundance levels in the majority of cases.

The positive performance of our metabolite prediction analyses within CCLE, between CCLE and NCI-60, and in the Aging patient dataset are based on the underlying metabolite/transcript correlation structure that can be calculated from the available data, and not on peculiarities of the chosen network inference algorithm, *corto*. In order to prove this, we repeated the analysis of CCLE-based prediction of CCLE metabolites, using 1000 splits of CCLE into training (50% of samples) and testing (50% of samples) parts. Our analysis shows that using real networks provides a significantly higher (*p* < 1 × 10^−302^) correlation between predicted and measured metabolite levels than shuffled networks, i.e., networks having the same structure as the inferred ones, but with transcript node labels shuffled, with the probability of each node to appear proportional to the degree of the node. While real networks provide an average predicted vs. real correlation coefficient of 0.424 (Figure 2C and Figure 5D), predictions do not correlate at all with experimental values when using shuffled networks (mean CC = 0.00, Figure 5D).

## 3. Discussion

In this study, we performed a proof-of-concept analysis to test whether metabolite levels could be predicted from transcript levels in cancer cells using a simple correlation-based metabolite/transcript network analysis. Our results indicate not only meaningful correlations between metabolites and genes (Figure 2A,B and Figure 4F and Appendix A), but also that such correlations can be used to predict metabolite levels within the largest dataset with metabolites and transcripts in the same cancer cell samples, the CCLE dataset (Figure 2D,E and Figure 3A,B). The prediction potential can be transferred to a different dataset, NCI-60 (Figure 3C,D), and even to patient-derived data (Figure 5B). It is also evident from our analysis that some metabolites, such as 1-methylnicotinamide, arginine and phosphocreatine (Figure 3A), can be predicted very well based on correlation with transcripts, while the prediction of a few metabolites cannot cross dataset boundaries (such as allantoin, Figure 3B,C). We also showed that the metabolite level prediction can be performed both on a sample-by-sample level (Figure 2D,E and Figure 3D) or on differential analysis scenarios (Figure 5B), and that it can be virtually applied to any sample with transcriptomics data available, as performed in the TCGA dataset, effectively transforming metabolite inferred abundances in derived data tracks for survival analysis (Figure 4A).

We hope that these preliminary results, in addition to discoveries previously made in simpler organisms [27], will aid in achieving a higher understanding of the metabolome–transcriptome interaction in cancer and pave the way for future metabolite quantitative predictors and novel tools for cancer diagnostics and, hopefully, intervention. One of the current limitations of the corto algorithm used in this study is that the removal of indirect correlations is based on DPI for metabolite–metabolite–transcript triplets, thereby making it impossible to remove higher order structures of indirect edges. The principle could be extended, for example, by adopting full partial correlation methods to remove more indirect correlations [46], and even give causal directionality to a subset of edges, thereby improving the accuracy and interpretability of the derived metabolite/transcript networks. The inferred networks by themselves stand as maps of functional relationships between metabolites and transcripts that go beyond the primary metabolic pathways involved, which is highlighted by the presence of non-metabolic genes in all the networks generated in this study (see Figure 4F, and Appendix A), and may reflect higher order molecular states of the cell.

Thus, in the networks inferred, the levels of the metabolites may reflect not only the activity of the genes encoding components of their metabolic pathways (which is often the case for the top correlations), but also an overall state of the cell. This could be illustrated by referring Figure 4F, which contains non-metabolic genes.

Future steps should involve ad-hoc experimental analysis to further prove the existence of transcript–metabolite correlations, and in which specific cancer contexts and for which specific metabolites it can be efficiently and robustly modeled, paired with the development of ad hoc algorithms. This will allow the possibility of a single diagnostic step to characterize, in a personalized manner, the transcript abundances of a specific cancer sample (via RNA-Seq) and then infer metabolomics patterns from transcriptional data. More future steps will also involve the analysis of whether and how metabolite/transcript correlation networks are conserved between physiological and pathological tissues, between tissues, and across species, to test the evolutionary conservation of the molecular relationships between metabolome and transcriptome.

## 4. Materials and Methods

### 4.1. CCLE Dataset

The CCLE cancer cell line dataset [19], comprising 890 samples with shared metabolomics and transcriptomics measurement, was obtained from the Broad Institute’s CCLE data portal (https://sites.broadinstitute.org/ccle/datasets, accessed on 20 March 2022).

### 4.2. NCI60 Dataset

The NCI-60 dataset was described and obtained from Siddiqui and colleagues [36] and comprised 57 samples with co-measured metabolites and gene expression levels.

### 4.3. TCGA Dataset

We downloaded a portion of the TCGA dataset, encompassing 14 tumor types, gene expression data, survival data, and proteomics data, from the Broad Institute GDAC portal (https://gdac.broadinstitute.org/, accessed on 20 March 2022) as described before [26]. RNA-Seq gene expression data was VST-normalized before the survival analysis [47].

### 4.4. Aging Dataset

Co-measured metabolites and transcript were available from human monocytes from old and young individuals in the dataset of Shchukina and colleagues [44], available at http://artyomovlab.wustl.edu/aging/download_data.html (accessed on 20 March 2022). The samples used for the old vs. young comparison were OD16 + OD18 and YD16 + YD21, respectively.

### 4.5. Network Generation

Metabolite-transcript networks were generated using the *corto* algorithm [28]. In brief, for gene expression, RNA-Seq data are normalized with Variance Stabilizing Transformation and microarray data with RMA transformation [48]. For metabolite data, scaled arbitrary units as provided from the original publications [24,36,44] were used.

A network is built using the combined metabolite and transcript data into a single matrix, specifying only metabolites as “centroids” (i.e., hubs in the resulting co-occurrence network [28]). Minimum Pearson’s correlation coefficient *p*-value to deem a putative edge significant was set to 0.001. The significance of each edge is tested via DPI and by 100 bootstraps. The following R code snippet describes the run, having “metabexpmat” as the combined input matrix, and “metabs” as a vector containing metabolite names:
library (corto)network ← corto (metabexpmat, centroids = metabs, nbootstraps = 100, *p* = 0.001)

### 4.6. Metabolite Prediction

Once a weighted co-occurrence network between metabolites and transcripts is defined, it can be applied to a different context where only transcripts have been measured, to infer metabolite abundances. Such abundances are expressed as Normalized Enrichment Scores [32] relative to either the mean of the dataset (for sample-by-sample inference) or for a contrast signature (e.g., in a comparison between two sample groups).

The skeleton R code to perform such an inference relies on the corto mra() function. For sample-by sample inference, assuming as “newexpmat” a second gene expression matrix, and as “network” the previous generated metabolite-transcript network:
sample_by_sample_prediction ← mra (newexpmat, regulon = network)

In addition, for a prediction of relative abundance between two sample groups, assume as “group1” and “group2” the expression matrices to compare:
contrast_prediction ← mra(group1, group2, regulon = network)

In the first case, a matrix providing predicted NES for each metabolite (rows) and each sample (column) is provided. In the second case, a vector with NESs and *p*-values relative to the predicted differential abundance of every metabolite is provided. A more in-depth description of the corto functions used in this study is available on the corto Github page (https://github.com/federicogiorgi/corto, accessed on 28 March 2022).

### 4.7. Additional Analyses

Every provided graphical operation and statistical test were performed using the R software v4.1.1. Prominent packages used were corto [28], ggrepel, VennDiagram [49], and DESeq2 for RNA-Seq data normalization [47]. Survival analysis was performed as in [50], leveraging the R survival package and the coxph() function for the calculation of the Wald test and the Hazard Ratio (HR). All *p*-values are corrected for false discovery rate using the Benjamini–Hochberg method.

## Figures and Tables

**Figure 1 ijms-23-03867-f001:**
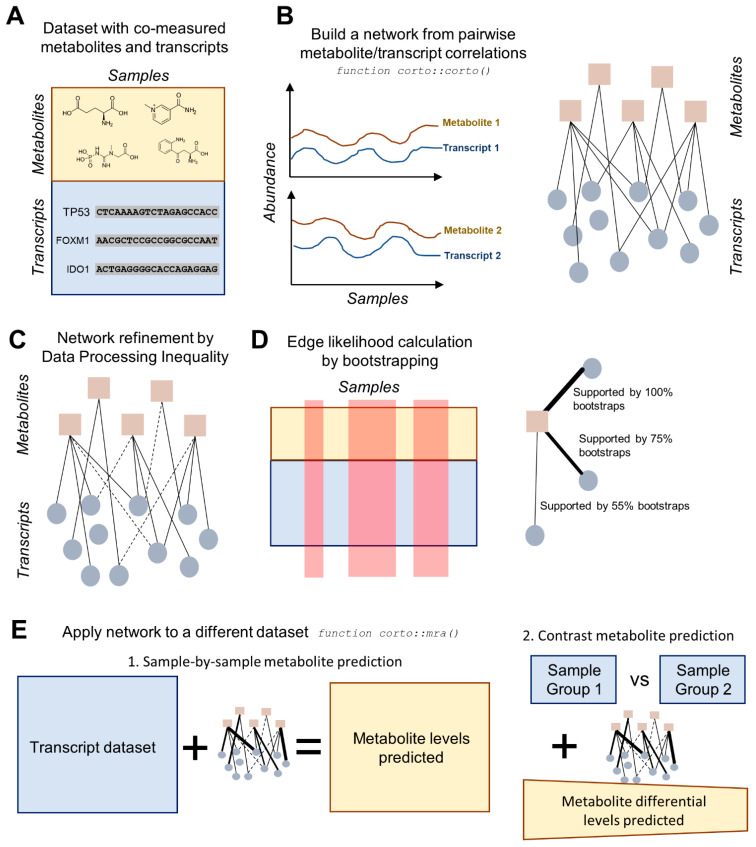
Diagram highlighting the main steps of the transcript-based metabolite prediction approach. (**A**) The starting point must be a dataset with metabolites and transcripts co-measured across several samples. Ideally, transcriptome-wide (>20k transcripts) measurements should be obtained in a large dataset (>50 samples). **(B**) pairwise Pearson correlation coefficients are calculated across the dataset for every metabolite/transcript pair, resulting in a metabolite/transcript network; (**C**) the network is processed according to Data Processing Inequality (DPI), which removes indirect correlations; (**D**) edge likelihood is calculated through a bootstrapping process of the initial dataset, providing the number of bootstraps supporting a specific edge. Steps from B to D are executed using the corto() function from the R package *corto*; (**E**) the network, with indirect edges removed and all remaining edges weighted by both correlation coefficient and likelihood, is applied to a second transcript dataset, in order to obtain predicted metabolite levels. This can be applied on a sample-by-sample basis, yielding predicted metabolite levels for each sample, or on a contrast basis, providing the predicted differential levels between two groups of samples (e.g., treated vs. control). The application of a network to a second dataset is performed by the mra() function of the *corto* package.

**Figure 2 ijms-23-03867-f002:**
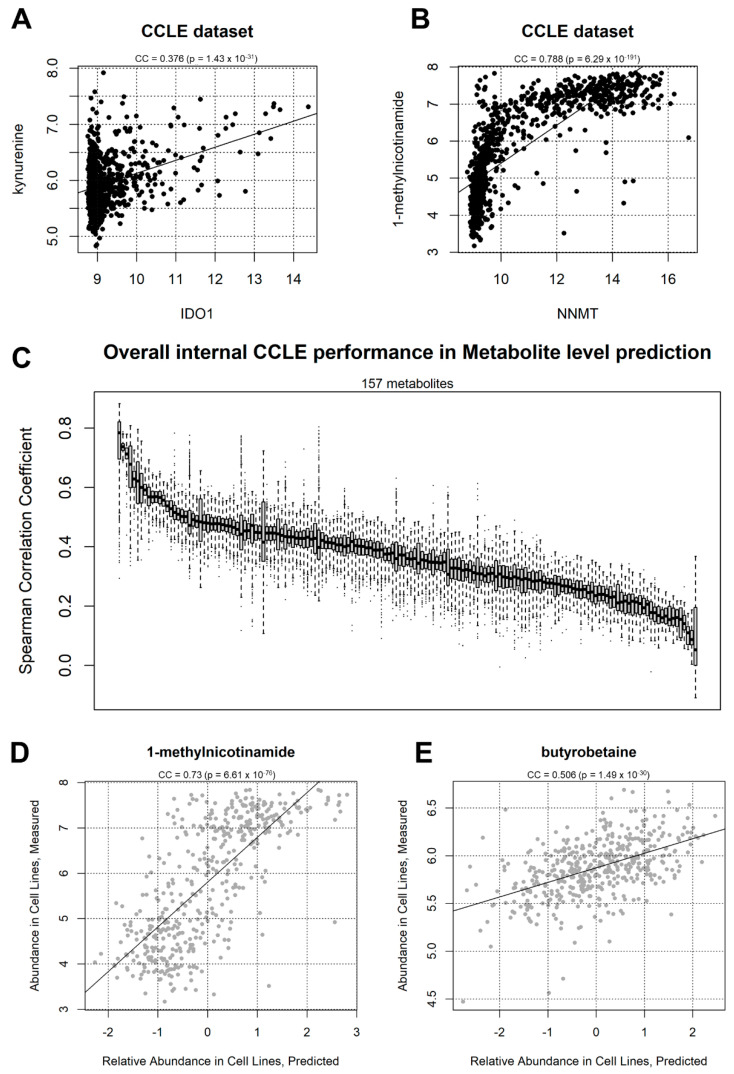
(**A**) Correlation between *IDO1* gene expression and kynurenine abundance in the CCLE dataset. The Coefficient of Correlation (CC) and the associated *p*-value are indicated. (**B**) correlation between *NNMT* gene expression and 1-methylnicotinamed abundance in the CCLE dataset; (**C**) overall correlation coefficient distribution between real and predicted metabolite abundance value in 157 metabolites. Each boxplot represents a metabolite, sorted by average correlation, tested across 2000 CCLE partitions (50% of the data used as training and 50% as testing). (**D**) example correlation between 1-methylnicotinamide measured levels (*y*-axis) and predicted levels according to *corto*-aggregated gene expression profiles (*x*-axis). Each point represents a CCLE sample taken from 445 samples (50%) not used for metabolite-transcript network generation. The example represents an arbitrary split of the dataset into training and testing samples, providing a correlation coefficient similar to the average deducted from 2000 samplings for 1-methylnicotinamide, shown in the previous panel; (**E**) another correlation example between measured (*y*-axis) and predicted (*x*-axis) butyrobetaine levels.

**Figure 3 ijms-23-03867-f003:**
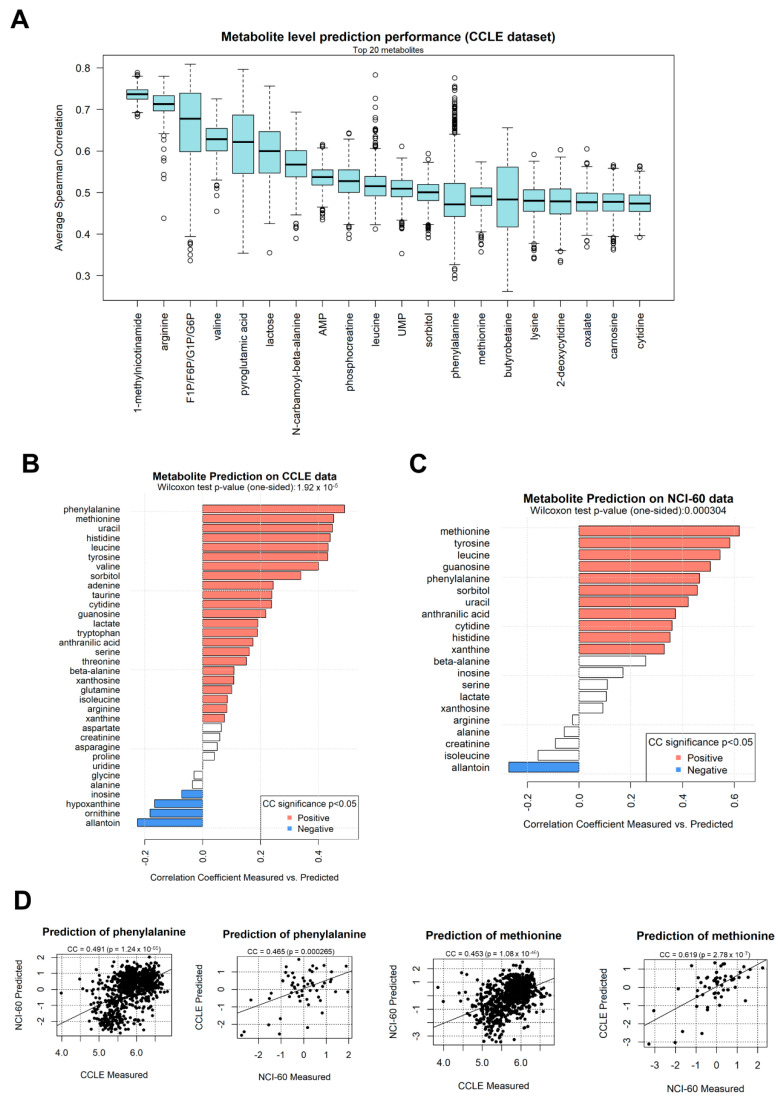
(**A**) Correlation coefficient distribution between predicted and measured metabolite levels within the CCLE dataset, using 2000 training/test splits each taking separate halves of the full dataset. This panel shows the 20 polar metabolites with higher correlation coefficients and can effectively be considered as a labeled section of Figure 2C; (**B**) correlation coefficients calculated between predicted and experimentally measured metabolite levels in the CCLE dataset, using a network model extracted from the NCI-60 dataset. Bars are colored in red if the correlation coefficient is significant (*p* < 0.05) and positive, in blue if significant and negative, in white otherwise; (**C**) correlation coefficients calculated between predicted and experimentally measured metabolite levels in the NCI-60 dataset, using a network model extracted from the CCLE dataset; (**D**) scatter plots depicting selected examples of metabolites measured in dataset A (x-axis, NCI-60 or CCLE) and predicted using a network based on dataset B (*y*-axis, CCLE or NCI-60).

**Figure 4 ijms-23-03867-f004:**
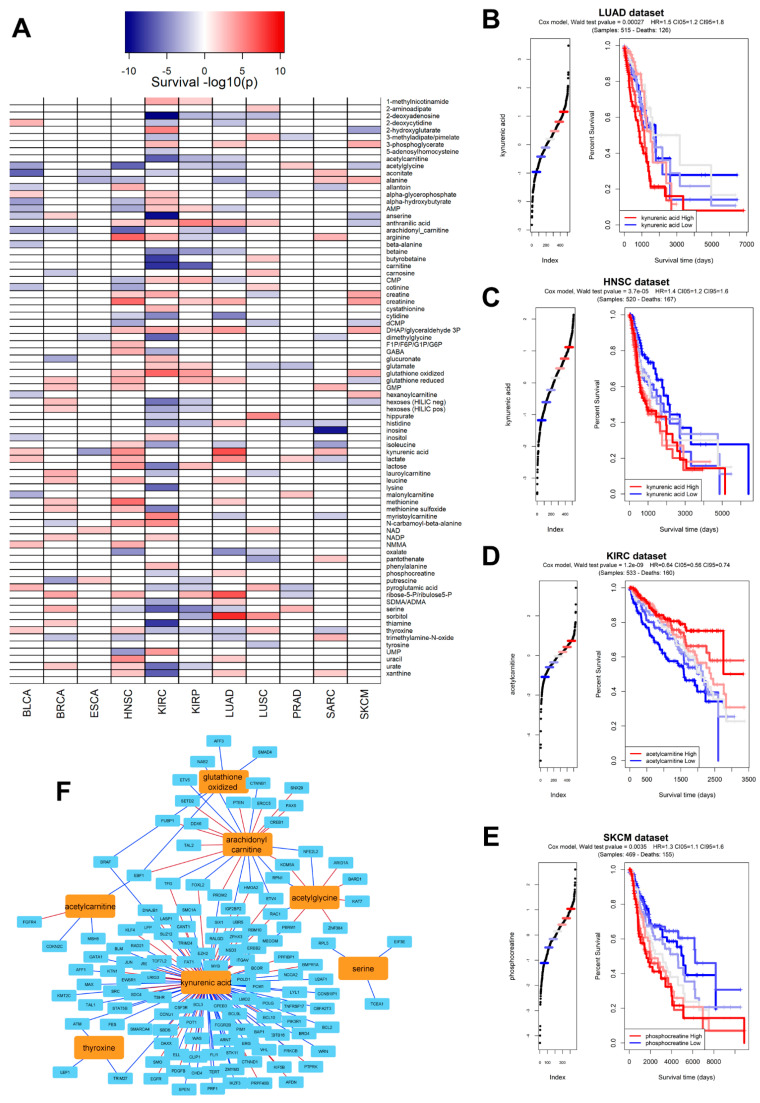
(**A**) Heatmap indicating the significance in survival prediction for all polar metabolite levels predicted in the TCGA dataset using the CCLE-based metabolite-transcript network; metabolites are depicted as rows and TCGA datasets as columns. Metabolite profiles not significantly associated (*p* > 0.01) to survival are shown in white color Metabolites whose higher abundance is significantly associated to worse survival are depicted in red. Metabolites whose higher abundance is significantly associated to better survival are depicted in blue. Colors are proportional to −log_10_(p_survival_) as indicated by the key above the heatmap; (**B**) Kaplan–Meier (KM) graph recapitulating a survival analysis calculated by segregating patients in the LUAD (Lung Adenocarcinoma) TCGA dataset according to predicted levels of kynurenic acid. Reported, the Wald test from a Cox model, the Hazard Ratio (HR), and the Confidence Intervals (CI) of the HR at different thresholds (5% and 95%). (**C**) KM graph recapitulating a survival analysis calculated by segregating patients in the HNSC (Head and Neck Squamous carcinoma) TCGA dataset according to predicted levels of acetylcarnitine; (**D**) KM graph recapitulating a survival analysis calculated by segregating patients in the KIRC (Kidney Renal Clear cell Carcinoma) TCGA dataset according to predicted levels of acetylcarnitine; (**E**) KM graph recapitulating a survival analysis calculated by segregating patients in the SKCM (Skin Cutaneous Melanoma) TCGA dataset according to predicted levels of phosphocreatinine; (**F**) visual representation of the *corto*-inferred network including seven metabolites (in orange) most associated with survival prediction, calculated as the sum of all −log_10_(p_survival_) across the TCGA dataset and genes (in light blue) derived from the COSMIC Cancer Gene Census [43]. Red edges depict significant positive correlations, while blue edges depict significant negative correlations.

**Figure 5 ijms-23-03867-f005:**
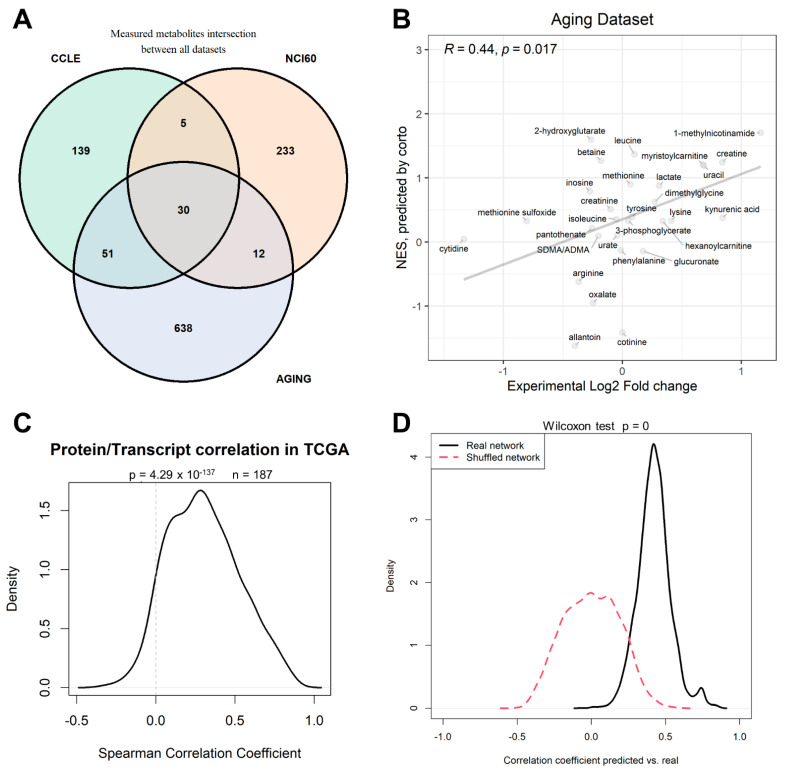
(**A**) Venn diagram depicting the number of metabolites measured in the three datasets analyzed in this paper: CCLE, NCI-60, and aging [44] datasets; (**B**) correlation in the monocyte patient dataset from the aging dataset between calculated differential abundance (*x*-axis) expressed as Log_2_ Fold Change of old vs. young samples, and inferred differential abundance (*y*-axis) expressed as *corto*-calculated NES (Normalized Enrichment Score) using the CCLE-based metabolite-transcript network (see Appendix A) on gene expression profiles. (**C**) distribution of correlation coefficients between protein and transcript abundances from the same genes measured in TCGA datasets. (**D**) Distribution of correlation coefficients of predicted vs. measured metabolite levels in the CCLE dataset using 50%/50% testing/training split prediction across 1000 CCLE dataset splits; networks used in the prediction were correctly inferred (black solid line) or shuffled (red dashed line).

## Data Availability

The data for our paper is available at the following online public repositories:
CCLE: https://sites.broadinstitute.org/ccle/datasetsNCI-60: https://github.com/Mathelab/NCI60_GeneMetabolite_DataTCGA: https://gdac.broadinstitute.org/Aging dataset: http://artyomovlab.wustl.edu/aging/download_data.html. CCLE: https://sites.broadinstitute.org/ccle/datasets NCI-60: https://github.com/Mathelab/NCI60_GeneMetabolite_Data TCGA: https://gdac.broadinstitute.org/ Aging dataset: http://artyomovlab.wustl.edu/aging/download_data.html.

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
