# Peer review of "Prediction of Metabolic Profiles from Transcriptomics Data in Human Cancer Cell Lines"

_ijms, 2022, doi:10.3390/ijms23073867_

Round 1

Reviewer 1 Report

The authors described a metabolite/transcript correlation network that is used to predict metabolite levels in cancer cell line dataset and TCGA dataset. Several major points need to be addressed.

  1. A flow of diagram of this study can largely improve the readability.
  2. Figure 1A: The rationale behind choosing IDO/kynurenine should be specified.
  3. Figure 3: The low resolution hinders the legibility. Consider improving the Figure 3. What is the hazard ratio (HR) and logrank p-value?

Reviewer 2 Report

The manuscript is acceptable for publication. No further comments.

Reviewer 3 Report

Cavicchioli et al. take advantage of the elegant algorithm using partial correlations and Data Processing Inequality (DPI) implemented in their published program corto, and leverage on the availability of large cancer cell line datasets that contain the transcriptome and the metabolome information on the same samples, to show that it is possible to use correlation structures between the transcriptome and the metabolome to predict relative levels of multiple (but not all) metabolites starting from the transcriptome. The Authors demonstrate successful predictions between independent partitions (halves) of the the same cancer cell line dataset. They follow to demonstrate predictions between these two cancer cell line datasets, and also from a cancer cell line dataset to in vivo data from cancer or a specific non-cancer cell type (monocyte) in normal development (aging).

The results are convincing and well-presented. However, some aspects of the methodology need to be better described, a limitation of the used algorithm – as applied in the current work – should me mentioned, and the discussion of the biological interpretation of the results could be improved before this work is ready for publication. Specifically:

1) In the current work, the Authors use two functions: corto – for network inference, and mra – for prediction. While the algorithm used by the corto function is adequately described on the program Github site, provided in the cited ref. [28], the operation of the mra function is not. A brief description of mra operation should be given, either in the supplement to the current work or on the corto Github site.

2) On page 6 the Authors write: “using the corto algorithm, which in brief calculates all pairwise correlations and removes indirect ones by virtue of applying DPI [28].” This should be made more precise, e.g. by changing to: “removes indirect ones through another metabolite by virtue of applying DPI [28]. Given the option “centroids=metabs” with which the corto was run and the description of the corto function on the program Github site, the DPI is applied to all to centroid-centroid-target triplets. Therefore, it can only remove indirect correlations between a metabolite and a target (transcript) through another metabolite, however it cannot remove an indirect correlation between a metabolite and a transcript through another transcript.

3) The above shows a (minor) limitation of the current use of the corto algorithm, which should be mentioned in the Discussion, namely that that the inferred networks contain indirect relationships through different transcripts to a given metabolite. Thus, in the networks inferred, the levels of the metabolites may reflect not only the activity of the genes encoding components of their metabolic pathways (which is often the case for the top correlations), but also an overall state of the cell. This could be illustrated by referring Figure 3 F, which contains non-metabolic genes.

Round 2

Reviewer 1 Report

Most of the issues have been addressed except for the Figure 4, in which the letters in Figure B, C, D, and E are indistinct.